# Atrial fibrillation as a novel risk factor for retinal stroke: A protocol for a population-based retrospective cohort study

Jay B. Lusk[1,2,3], Lauren Wilson[1], Vinit Nalwade[1], Ailin Song[4], Matthew Schrag[5], Valerie Biousse[6,7], Fan Li[8], Sven Poli[9], Jonathan Piccini[10,11], Ying Xian[12,13], Emily O'Brien[1,2], Brian Mac Grory[2,4]*

1 Department of Population Health Sciences, Duke University, Durham, North Carolina, United States of America, 2 Department of Neurology, Duke University School of Medicine, Durham, North Carolina, United States of America, 3 Duke University Fuqua School of Business, Durham, North Carolina, United States of America, 4 Department of Ophthalmology, Duke University School of Medicine, Durham, North Carolina, United States of America, 5 Department of Neurology, Vanderbilt University School of Medicine, Durham, North Carolina, United States of America, 6 Department of Neurology, Emory University School of Medicine, Durham, North Carolina, United States of America, 7 Department of Ophthalmology, Emory University School of Medicine, Atlanta, Georgia, United States of America, 8 Department of Biostatistics, Duke University School of Medicine, Durham, North Carolina, United States of America, 9 Department of Neurology & Stroke, University of Tübingen, Tübingen, Germany, 10 Duke Clinical Research Institute, Durham, North Carolina, United States of America, 11 Department of Medicine, Duke University School of Medicine, Durham, North Carolina, United States of America, 12 Department of Neurology, University of Texas- Southwestern Medical Center, Dallas, Texas, United States of America, 13 Department of Population Health Sciences, University of Texas-Southwestern Medical Center, Dallas, Texas, United States of America

* brian.macgrory@duke.edu

**Data Availability Statement:** No datasets were generated or analyzed during the current study. All relevant data from this study will be made available upon study completion upon execution of an

## Abstract

Central retinal artery occlusion (CRAO; retinal stroke or eye stroke) is an under-recognized, disabling form of acute ischemic stroke which causes severe visual loss in one eye. The classical risk factor for CRAO is ipsilateral carotid stenosis; however, nearly half of patients with CRAO do not have high-grade carotid stenosis, suggesting that other cardiovascular risk factors may exist for CRAO. Specifically, prior studies have suggested that cardioembolism, driven by underlying atrial fibrillation, may predispose patients to CRAO. We describe the design of an observational, population-based study in this protocol. We evaluate two specific objectives: 1) To determine if atrial fibrillation is an independent risk factor for CRAO after adjusting for medical and cardiovascular risk; 2) To determine if use of oral anticoagulation can modify the risk of CRAO for patients with atrial fibrillation. This protocol lays out our strategy for cohort definition, case and control definition, comorbidity ascertainment, and statistical methods.

## Introduction

### Significance/overall impact

Central retinal artery occlusion (CRAO; retinal stroke or eye stroke) is an under-recognized, disabling form of acute ischemic stroke which causes severe visual loss in one eye. Despite its

authorized Data Use Agreement from the Centers for Medicare and Medicaid Services.

**Funding:** This research was funded by the National Institutes of Health (K23HL161426) (BM). The funder did not and will not have a role in study design, data collection and analysis, decision to publish, or preparation of the manuscript.

**Competing interests:** "I have read the journal's policy and the authors of the manuscript have the following interests: J.B. Lusk is supported by the American Heart Association. Dr Biousse is supported by the National Institutes of Health National Eye Institute core grant P30-EY06360 (Department of Ophthalmology, Emory University School of Medicine) and by a departmental grant from the Research to Prevent Blindness (New York, NY) and is a consultant for GenSight Biologics and Neurophoenix. Dr Schrag reports compensation from the REVISION trial for data and safety monitoring services; compensation from Raymond James for consultant services; grants from the National Institutes of Health; and compensation from Labaton Sucharow for consultant services. Dr Poli reports grants from the Helena Laboratories Corporation, Bristol Myers Squibb, Boehringer Ingelheim, and the Daiichi Sankyo Company; compensation from Alexion Pharmaceuticals, Inc, Portola Pharmaceuticals LLC, Werfen USA LLC, and AstraZeneca for consultant services; and compensation from Boehringer Ingelheim, Bristol Myers Squibb, the Daiichi Sankyo Company, and Bayer Healthcare for other services. Dr Piccini reports compensation from Biotronik, Inc, Philips, Medtronic, Boston Scientific Corporation, ResMed Foundation, Ablacon, LivaNova USA, Inc, AltaThera Pharmaceuticals LLC, AbbVie Biotherapeutics, Abbott Laboratories, ElectroPhysiology Frontiers, Itamar Medical, Inc, ARCA biopharma, Pfizer, Bristol Myers Squibb, Sanofi US Services, Inc, and Milestone Scientific, for consultant services; employment by the Duke University; and grants from the American Heart Association and iRhythm Technologies. Dr O'Brien reports compensation from Boehringer Ingelheim for consultant services and employment by the Duke University. B. Mac Grory is supported by a grant from the National Institutes of Health (K23HL161426). The other authors report no conflicts. This does not alter our adherence to PLOS ONE policies on sharing data and materials."

formal recognition as a form of stroke by the American Heart Association/American Stroke Association (AHA/ASA), there are no guidelines in stroke medicine, cardiology, or neurology to inform its management [1]. In parallel, the National Heart, Lung, and Blood Institute has identified an unmet need to improve the identification of patients harboring atrial fibrillation (AF), many of whom remain undiagnosed. There are estimated to be approximately 700,000 such patients in the US [2]. CRAO represents an intriguing new dimension in the identification of patients harboring AF and therefore at risk of a future, lethal cardioembolic stroke. An AHA/ASA Scientific statement endorsed by 6 other professional organizations in the US recently called for urgent research on its mechanistic underpinnings [1]. Until this critical knowledge gap is filled, there may be a vulnerable population exposed to future cardioembolic stroke.

## Biological rationale

The prevailing paradigm is that CRAO arises because of ipsilateral stenotic carotid plaque [3] releasing predominantly calcium or cholesterol emboli. However, this observation is drawn from studies evaluating patients with branch retinal arterial occlusion (BRAO) where emboli may be visualized on funduscopic examination [4]. This may have falsely increased the perceived contribution of calcium/cholesterol emboli to CRAO as these emboli are less likely to autolyze when compared with fibrin-based emboli and are thus visible longer. Recently, we demonstrated that tissue plasminogen activator (tPA, a drug that breaks up fibrin-based clots) improved the rate of visual recovery in some patients with CRAO, providing evidence against the assertion that CRAO is caused exclusively by cholesterol or calcium emboli [5–7]. Furthermore, over 50% of people with CRAO do not have high-grade ipsilateral carotid stenosis [8] and 30% of people with CRAO do not have even mild ipsilateral carotid disease [9].

## Prior work addressing this question

There is a higher burden of AF in people with CRAO than in the general population. One prior insurance-based cohort study in Taiwan described a higher incidence of CRAO in patients with AF than in the general population [10]. This study adjusted for age, sex and year of presentation but did not adjust for vascular risk factors or medication exposures. Additionally, other studies have reported a higher prevalence of AF in patients with CRAO [11, 12] than in the general population which lends credence to the hypothesis that the two conditions are associated. However, CRAO (like cerebral stroke) is a surrogate marker for vascular risk [8] so studies that comprehensively address this confounding are needed. We performed a study using healthcare cost and utilization project (HCUP) data from New York, California, and Florida hospitalizations and emergency department visits found that there was an inverse relationship between AF and CRAO [13]. However, this study lacked access to outpatient and prescription drug information, which resulted in systematic under-ascertainment of both AF and CRAO. Furthermore, this study could not evaluate whether exposure to anticoagulation in patients with AF could prevent the development of CRAO. Therefore, a study using inpatient, outpatient, and pharmacy claims data would be essential to definitively determine whether cardioembolism is a potential causative etiology for CRAO, and whether anticoagulation can modify this risk factor. That only 50–60% of people meeting guideline criteria for anticoagulation for AF are actually treated with an anticoagulant creates an intriguing opportunity to leverage real-world data to conduct a quasi-randomized study to examine the interplay between anticoagulation and CRAO risk [14].

## Materials and methods

This study was deemed exempt by the Duke University Institutional Review Board, protocol number Pro00110768.

### Aims

Our aims are as follows:

1. Determine if prevalent AF is a risk factor for CRAO independent of other vascular risk factors

2. Determine if anticoagulant usage modifies the risk of CRAO among patients with AF by means of a quasi-randomized study focused on patients with established AF

### Design and setting

Retrospective, observational, population-level cohort study of patients with atrial fibrillation and matched controls using administrative claims data among a 5% sample of fee-for-service Medicare beneficiaries in the United States from 2000–2020. This data source allows us to follow unique patients longitudinally for the duration of their fee-for-service enrollment, regardless of state of residence.

### Sample size calculation

Power/sample size calculations were made on the basis of an unadjusted Cox proportional hazards regression model. Within the primary analysis of aim 1, assuming a loss to follow-up of 20%, 570,000 subjects (285,000 with AF and 285,000 without) would have 90% power to detect a hazard ratio of 2, at an alpha of 0.05. This conservatively assumes an event rate of 22/100,000 per year in the exposed group (a rate of 45/100,000 is reported previously) [10] and an event rate of 11/100,000 per year in the unexposed group (a rate comparable to that reported previously) [15]. A query of the Medicare 5% sample between 2000 and 2015 reveals 818,940 unique patients with AF and 11,400 unique patients with a new diagnosis of CRAO. Power calculations were performed with the *powerSurvEpi* package (R v4.0.3).

### Inclusion and exclusion criteria (Aim 1)

The inclusion criterion is continuous fee-for-service Medicare enrollment for at least 12 months prior to AF diagnosis or matched index date (July 1 of inclusion year) for controls and continuous enrollment in fee-for-service (FFS) Medicare after AF diagnosis/index date for at least 12 months or until death if death occurred before 12 months of continuous enrollment.

   Patients with prevalent AF will be identified as follows:ICD-9-CM code 427.31 or ICD-10-CM code I48 in 2 outpatient claims or 1 inpatient claim within 365 days, in any diagnosis position of the claim [16–18]. The index date will be the date of the first qualifying inpatient claim or second qualifying outpatient claim. Patients will be required to have 12 months of FFS claims without an AF diagnosis prior to the index AF claims based on inclusion criteria.

   Exclusion criteria are:

1. patients with any claim for CRAO in the 12 months prior to AF diagnosis or index date for controls (July 1 of potential inclusion year).

### Inclusion and exclusion criteria (Aim 2)

Inclusion criteria are fee-for-service Medicare enrollment for at least 12 consecutive month and 12 months FFS with Part A, B, and D enrollment following the index date for at least 12 months or until death, and presence of atrial fibrillation as defined in the exposure section. The index date for patients exposed to anticoagulation will be the date of the first anticoagulant prescription drug claim occurring after the diagnosis of AF. For potential controls, only patients with no exposure to anticoagulation throughout the study period would be eligible to avoid selecting controls who start an anticoagulant during the follow-up period. For controls, inclusion and exclusion criteria will be examined for each year after AF diagnosis, and the midpoint of one random year will be selected as the index date for each control patient.

<u>Exclusion criteria include</u>:

1. Patients with any claim for CRAO in the 12 months prior to AF diagnosis or index date for controls;

2. Patients who die within first 12 months of enrollment;

3. Patients taking therapeutic-dose heparin, enoxaparin, dalteparin, fondaparinux, or betrixaban.

### Exposure

The primary exposure of interest is prevalent atrial fibrillation/atrial flutter. Atrial fibrillation/ atrial flutter will be defined through ICD-9-CM 427.13 or ICD-10-CM code I48 in 2 outpatient claims or one inpatient claim within 365 days [16–18]. Episodes of atrial fibrillation within 1 month of inpatient surgery will be excluded to avoid including patients with isolated post-operative atrial fibrillation [19].

The secondary exposure of interest (among patients with atrial fibrillation) is exposure to anticoagulation. For our secondary analysis (Objective 2), we will utilize Medicare Part D pre-scription drug claims to identify patients taking a direct acting oral anticoagulant (DOAC) or warfarin.

A list of search terms to identify anticoagulant drugs will be utilized as per Table 1, below:

### Outcome

The outcome of interest is incident central retinal artery occlusion. Central retinal artery occlusion will be identified through ICD-9-CM 362.31, which we validated in a prior study [13] in the primary diagnosis position in any emergency department, inpatient, or outpatient

**Table 1. Drug lookup queries for anticoagulant exposure.**

| Drug Name | |
|---|---|
| Warfarin | Part D claim for generic drug name like '%WARFARIN%', summarized per month (90-day fills will be imputed to monthly indicators) |
| Apixaban | Part D claim for generic drug name like '%APIXABAN%', summarized per month (90-day fills will be imputed to monthly indicators) |
| Dabigatran | Part D claim for generic drug name like '%DABIGATRAN%', summarized per month (90-day fills will be imputed to monthly indicators) |
| Edoxaban | Part D claim for generic drug name like '%EDOXABAN%', summarized per month (90-day fills will be imputed to monthly indicators) |
| Rivaroxaban | Part D claim for generic drug name like '%RIVAROXABAN%', summarized per month (90-day fills will be imputed to monthly indicators) |

claim. We will use the corresponding ICD-10-CM code H34.1x for claims occurring after September 31, 2015.

## Adjustment covariates

We will utilize the following pre-specified covariates for risk adjustment: age, biological sex, race/ethnicity, and cardiovascular comorbidities and medical risk factors, including the medical comorbidities included in the Charlson Comorbidity Index (except for cardiac arrhythmias, which subsumes our primary exposure of interest), and additionally including hypertension, hyperlipidemia, valvular heart disease, and smoking status, using validated ICD-9-CM and ICD-10-CM algorithms [20–23]. Carotid stenosis and Stroke/TIA are contained in the cerebrovascular disease group of the Charlson Comorbidity Index. S1 and S2 Figs show the directional acyclic graphs representing our understanding of possible confounding variables.

We will ascertain comorbidities using a twelve-month look-back period from the patient's index date (July 1 of initial year of inclusion in cohort for controls) using established claims-based algorithms. These are outlined in Table 2, below.

## Analytical cohort

An analytic cohort will be generated by applying the below inclusion and exclusion criteria as follows:

All patients meeting the criteria for AF in the study period and meeting other inclusion/exclusion criteria will be identified as cases, and the date of AF diagnosis will serve as case's index date for calculation of covariates. All remaining beneficiaries will be considered potential controls; inclusion and exclusion criteria will be applied based on an index date of July 1 for each year in the study period; all beneficiaries meeting inclusion/exclusion criteria for each year will be included in the potential control pool. For all potential controls with multiple index dates/years of eligibility, the year they contribute will be selected randomly. A multivariable logistic regression analysis will be conducted using cases and potential controls to create propensity scores for AF status; the final control group will be selected using propensity score matching to the AF cases.

Survival time for patients will be defined as the number of months from the index date when criteria were met. Patients will be censored at death, at the end of fee-for-service enrollment, or at the end of data availability. When 85% of the cohort has died or been lost to follow-up, we will censor remaining patients at that point to avoid instability in estimates due to very small numbers of remaining participants.

Details of the cohort analysis will be reported using RECORD guidelines for analyses of routinely collected data, including use of a patient flow diagram [24].

## Statistical analysis plan

**Re-estimation of propensity scores and overlap weighting.** To adjust for the observed confounding and focus on the patient population in clinical equipoise, we will first re-estimate the propensity scores in the matched sample and then apply propensity score overlap weighting [25–27] to ensure exact balance of baseline covariates between patients with and without AF.

**Descriptive analysis.** Using the weighted analytic cohort, we will calculate means with standard deviations and medians with interquartile ranges to summarize the distribution of continuous, normally distributed variables, and medians with interquartile ranges to summarize the distribution of continuous, not normally distributed variables. We will summarize categorical data using counts with percentages. Data will be stratified by presence of atrial

**Table 2. List of international classification of diseases codes used for comorbidity adjustment.**

| Comorbidity | ICD-9 Code | ICD-10 Code | Citation |
|---|---|---|---|
| Hypertension | 401.*, 402.*, 403.*, 404.*, 405.*, 437.2* | I10.*-I13.*, I15.0, I15.2, I15.8, I15.9, I16.*, I67.4* | Birman-Deych, 2005 [21] |
| Hyperlipidemia† | 272.x | E78.x | Oake, 2017 [22] |
| Tobacco Use†† | 305.1, 649.0x, 989.84, V15.82; CPT codes 99406, 99407, G0436, G0437, G9016, S9453, S4995, G9276, G9458, 1034 F 4004 F 4001 F | F17.x, T65.2xx, Z87.891, CPT codes 99406, 99407, G0436, G0437, G9016, S9453, S4995, G9276, G9458, 1034 F 4004 F 4001 F | Desai, 2016 [23] |
| Valvular Heart Disease | 394.*, 395.*, 396.*, 397.*, 398.9*, V42.2*, V43.3*, 424.* | I05.*-I08.*, I09.1*, I09.81, I09.89, I09.9*, I34.*-I39.*, Z95.2*, Z95.3*, Z95.4* | Birman-Deych, 2005 [21] |
| **Charlson Comorbidity Variables** | | | |
| Myocardial Infarction | 410.*, 412.* | I21.*, I22.*, I25.2* | Quan, 2005 [20] |
| Congestive Heart Failure | 398.91, 402.01, 402.11, 402.91, 404.01, 404.03, 404.11, 404.13, 404.91, 404.93, 425.4, 425.5, 425.7, 425.8, 425.9, 428* | I09.81, I11.0*, I13.0*, I13.2*, I42.0*, I42.5*-I42.9*, I43.*, I50.* | Quan, 2005 [20] |
| Peripheral Vascular Disease | 093.0*, 437.3*, 440.*, 441.*, 443.1*, 443.2*, 443.8*, 443.9*, 447.1*, 557.1*, 557.9*, V43.4* | A52.01*, E08.51, E08.52, E09.51, E09.52, E10.51, E10.52, E11.51, E11.52, E13.51, E13.52, I67.0*, I67.1*, I70.*, I71.*, I73.1*, I73.8*, I73.9*, I77.7*, I79.*, K55.1*, K55.8*, K55.9*, Z958.2* | Quan, 2005 [20] |
| Cerebrovascular Disease††† | 430*, 431*, 432.*, 433.*, 434.*, 435.*, 436*, 437.*, 438.* | G45.0, G45.1, G45.2, G45.4, G45.8, G45.9, G46.*, I60.*-I63.*, I65.*-I66.*, I67.1*, I67.2*, I67.4*, I67.5*, I67.6*-I67.7*, I67.81, I67.82, I67.84, I67.89, I67.9*, I68.*-I69.* | Quan, 2005 [20] |
| Dementia | 290.*, 294.1*, 331.2* | F01.*, F02.*, F03.9*, G31.1* | Quan, 2005 [20] |
| Chronic Pulmonary Disease | 416.8*, 416.9*, 490*, 491.*, 492.*, 493.*, 494.*, 495.*, 496*, 500*, 501*, 502*, 503*, 504*, 505*, 506.4*, 508.1*, 508.8* | I27.2*, I27.81, I27.89, I27.9*, J40.*-J44.*, J45.2*-J45.5*, J45.90, J45.99, J47.*, J60.*, J61.*-J67, J68.4*, J70.1*, J70.2*, J70.3*, J70.4*, J70.8* | Quan, 2005 [20] |
| Connective Tissue Disease | 446.5*, 710.0*, 710.1*, 710.2*, 710.3*, 710.4*, 714.0*, 714.1*, 714.2*, 714.8*, 725* | M05.*, M06.*, M31.5*, M31.6*, M32.*-M34.*, M35.0*, M35.3*, M36.0* | Quan, 2005 [20] |
| Peptic Ulcer Disease | 531.*, 532.*, 533.*, 534.* | K25.*-K28.* | Quan, 2005 [20] |
| Mild Liver Disease | 070.22, 070.23, 070.32, 070.33, 070.44, 070.54, 070.6*, 070.9*, 570*, 571.*, 573.3*, 573.4*, 573.8*, 573.9*, V42.7* | B17.9*, B18.0*-B18.2*, B19.0*, B19.9*, K70.0, K70.1*, K70.2, K70.3*, K70.40, K70.9, K71.*, K72.00, K73.*, K74.0*, K74.1*-K74.6*, K75.2*-K75.4*, K75.8*, K75.9*, K76.0*-K76.5*, K76.89, K76.9*, K77.*, Z48.23, Z94.4* | Quan, 2005 [20] |
| Diabetes without complications | 250.0*, 250.1*, 250.2*, 250.3*, 250.8*, 250.9* | E10.1*, E10.618, E10.62*, E10.63, E10.64, E10.65, E10.69, E10.8*, E10.9*, E11.0*, E11.1*, E11.618, E11.62-E11.65, E11.69, E11.8*, E11.9*, E13.00, E13.01, E13.10, E13.11, E13.618, E13.62-E13.65, E13.69, E13.8*, E13.9* | Quan, 2005 [20] |
| Diabetes with complication | 250.4*, 250.5*, 250.6*, 250.7* | E10.2*-E10.5*, E10.610, E11.2*-E11.5*, E11.610, E13.2*-E13.5*, E13.610 | Quan, 2005 [20] |
| Paraplegia and Hemiplegia | 334.1*, 342.*, 343.*, 344.0*, 344.1*, 344.2*, 344.3*, 344.4*, 344.5*, 344.6*, 344.9* | G04.1*, G11.4*, G80.*-G82.*, G83.0*-G83.4*, G83.9* | Quan, 2005 [20] |
| Renal Disease | 403.01, 403.11, 403.91, 404.02, 404.03, 404.12, 404.13, 404.92, 404.93, 582.*, 583.0*, 583.1*, 583.2*, 583.4*, 583.6*, 583.7*, 585.*, 586*, 588.0*, V42.0*, V45.1*, V56.* | I12.0, I13.11, I13.2*, N03.*, N05.2*-N05.5*, N05.9*, N06.2*-N06.5*, N07.2*-N07.5*, N08.*, N17.1*, N17.2*, N18.*, N19.*, N25.0*, Z48.22, Z49.0*, Z49.3*, Z94.0*, Z99.2, Z91.15 | Quan, 2005 [20] |
| Non-metastatic cancer | 140.*, 141.*, 142.*, 143.*, 144.*, 145.*, 146.*, 147.*, 148.*, 149.*, 150.*, 151.*, 152.*, 153.*, 154.*, 155.*, 156.*, 157.*, 158.*, 159.*, 160.*, 161.*, 162.*, 163.*, 164.*, 165.*, 170.*, 171.*, 172.*, 174.*, 175.*, 176.*, 179.*, 180.*, 181.*, 182.*, 183.*, 184.*, 185.*, 186.*, 187.*, 188.*, 189.*, 190.*, 191.*, 192.*, 193.*, 194.*, 195.*, 200.*, 201.*, 202.*, 203.*, 204.*, 205.*, 206.*, 207.*, 208.*, 238.6* | C00.*-C26.*, C30.*-C34.*, C37.*-C41.*, C43.*, C45.0*-C45.2*, C45.7*, C45.9, C46.*-C58.*, C60.*, C61.*, C62.0*, C62.1*, C62.9*, C63.*-C76.*, C81.*-C86.*, C88.2*-C88.4*, C88.8*, C88.9*, C90.*-C93.*, C94.0*, C94.1*, C94.2*, C94.3*, C94.8*, C95.*, C96.0*, C96.2*, C96.4*, C96.9*, C96.A, C96.Z, D03.*, D45.*, D47.Z9 | Quan, 2005 [20] |
| Moderate to severe liver disease | 456.0*, 456.1*, 456.2*, 572.2*, 572.3*, 572.4*, 572.8* | I85.*, K70.41, K71.11, K72.01, K72.1*, K72.9*, K76.6*, K76.7* | Quan, 2005 [20] |

*(Continued)*

**Table 2.** (Continued)

| Comorbidity | ICD-9 Code | ICD-10 Code | Citation |
|---|---|---|---|
| Metastatic cancer | 196.*, 197.*, 198.*, 199.* | C77.*-C80.* | Quan, 2005 [20] |
| HIV/AIDS | 042*, 043*, 044* | B20.* | Quan, 2005 [20] |
| Depression | 296.2*, 296.3*, 296.5*, 300.4*, 309.*, 311* | F31.3*-F31.5*, F31.75, F31.76, F32.*, F33.*, F34.1*, F43.10, F43.11, F43.12, F43.2*, F43.8*, F43.9*, F93.0*, F94.8*); | Quan, 2005 [20] |
| Drug abuse | 292.*, 304.*, 305.2*, 305.3*, 305.4*, 305.5*, 305.6*, 305.7*, 305.8*, 305.9*, V65.42 | F11.*-F16.*, F18.*, F19.*, Z71.41, Z71.6, Z71.51 | Quan, 2005 [20] |
| Psychoses | 293.8*, 295.*, 296.04, 296.14, 296.44, 296.54, 297.*, 298.* | F06.*, F20.*, F22.*-F25.*, F28.*, F29.*, F30.2*, F31.2*, F31.5*, F32.3*, F33.3*, F44.89, F53.* | Quan, 2005 [20] |

[†]Claims data do not sensitively identify patients with hyperlipidemia. One prior validation study showed that this code was 32% sensitive/100% specific for hyperlipidemia [22].

[††]Claims data do not sensitively identify tobacco use, though are highly specific [23].

[†††] The Quan, 2005 algorithm uses ICD-9-CM 362.34 (transient retinal artery occlusion) and ICD-10-CM H34.0 (transient retinal artery occlusion) as part of the algorithm for identifying cerebrovascular disease; we have removed these specific codes from our comorbidity list due to the possibility that CRAO may have been miscoded as 362.34 or H34.0 due to similar clinical presentation.

fibrillation and include key demographic and clinical factors. For Aim 2, we will additionally summarize anticoagulant usage patterns among patients with atrial fibrillation, including class of agent utilized (DOAC vs. warfarin), number of monthly fills per calendar year, and duration of exposure in months. Table shells are provided in Tables 3–5.

**Univariable analysis.** We will calculate standardized differences for the variables of interest for the patient population with and without atrial fibrillation.

**Regression modeling (aim 1).**

a) The study index date will be the date of the first AF claim in the study and July 1 in the first year of inclusion in the analytic cohort for controls for the purposes of our survival analysis.

**Table 3. Characteristics of the overlap-weighted primary analysis cohort defined at first date of meeting study eligibility.**

| Characteristic | Total Sample (N = xxxx) | Patients with AF (n = xxxx) | Patients without AF (n = xxxx) | Standardized Difference |
|---|---|---|---|---|
| Age, mean (SD) | | | | |
| Women, n (%) | | | | |
| Race/ethnicity, n(%) | | | | |
| Asian/Pacific Islander | | | | |
| Black, non-Hispanic | | | | |
| Hispanic | | | | |
| White, non-Hispanic | | | | |
| Comorbidities (n, %) | | | | |
| Cerebrovascular Disease | | | | |
| Chronic Kidney Disease | | | | |
| Congestive Heart Failure | | | | |
| Coronary Artery Disease | | | | |
| Diabetes Mellitus | | | | |
| Hyperlipidemia | | | | |
| Hypertension | | | | |
| Peripheral Vascular Disease | | | | |
| Smoking | | | | |

**Table 4. Characteristics of the secondary analysis (Part-D enrolled) cohort defined at first date of meeting study eligibility.**

| Characteristic | Total Sample (N = xxxx) | Patients with at least one fill for anticoagulant (n = xxxx) | Patients without any anticoagulant use (n = xxxx) | Standardized Difference |
|---|---|---|---|---|
| Age, mean (SD) | | | | |
| Women, n (%) | | | | |
| Race/ethnicity, n(%) | | | | |
| Asian/Pacific Islander | | | | |
| Black, non-Hispanic | | | | |
| Hispanic | | | | |
| White, non-Hispanic | | | | |
| Comorbidities (n, %) | | | | |
| Cerebrovascular Disease | | | | |
| Chronic Kidney Disease | | | | |
| Congestive Heart Failure | | | | |
| Coronary Artery Disease | | | | |
| Diabetes Mellitus | | | | |
| Hyperlipidemia | | | | |
| Hypertension | | | | |
| Peripheral Vascular Disease | | | | |
| Smoking | | | | |
| Antiplatelet agent use | | | | |
| Oral anticoagulant use | | | | |
| Lipid-lowering agent | | | | |
| Oral hypoglycemic agent | | | | |
| Oral antihypertensive agent | | | | |

b) We will describe cumulative incidence of CRAO over time with the overlap-weighted cumulative incidence function and presented graphically for patients with and without atrial fibrillation. Next, a weighted, unadjusted cause-specific hazard model will be fit with AF as the univariable predictor and CRAO as the event of interest, with patients

**Table 5. Anticoagulation usage patterns among patients with atrial fibrillation with at least one Part D fill for an anticoagulant medication.**

| Characteristic | Total Sample (N = xxxx) |
|---|---|
| Number of monthly fills per calendar year (mean, SD) | |
| Class of Anticoagulant Agent | |
| DOAC | |
| Warfarin | |
| Specific Anticoagulant Drug Used | |
| Apixaban | |
| Dabigatran | |
| Edoxaban | |
| Rivaroxaban | |
| Warfarin | |
| Duration of Anticoagulant Exposure (months, mean, SD) | |
| Concurrent use of antiplatelet agent (count, %) | |

censored at death, at end of data availability, or at end of fee-for-service Medicare enrollment.

c) Then, a weighted cause-specific hazard model, clustered by state, and adjusting for age, sex, race, cardiovascular comorbidities will be fitted. Directly adjusted cumulative incidence curves for AF and control patients will be created with adjustment for all model covariates [28].

d) We will test the proportional hazards assumption for all models.

e) Our primary analytical model will be an overlap-weighted model with propensity scores derived from the adjustment covariates previously described as predictors for the presence of atrial fibrillation. We will report our results with sequential adjustment to produce the fully-adjusted, weighted model. We employ this sequential adjustment strategy to provide insight as to sources of measured confounding.

f) We will evaluate **positive control endpoints**. Our positive control endpoints will be any ischemic stroke (defined by ICD-9-CM 434.0, 434.00, 434.10, 434.11, 434.9, 434.90, 434.91, 436/ ICD-10-CM I66.09, I66.19, I66.29, I66.9, I63.40, I63.50, I67.89 in the primary diagnosis position in a hospitalization or ED visit) [29]. We will utilize the same modeling strategy as used for the primary outcome.

g) We will evaluate **negative control (falsification) endpoints**. Our negative control endpoint will be hospitalized/ED urinary tract infection (including renal abscess and pyelonephritis) (ICD-9-CM 590.1, 590.11 590.2, 590.8 or 599.0/ ICD-10-CM N15.1, N39.0, N10). We will utilize the same modeling strategy as used for the primary outcome [30].

h) We will perform several sensitivity analyses for outcome ascertainment.

1) We will repeat the primary analysis, classifying CRAO based on the presence of the ICD codes in any diagnosis position rather than the primary diagnosis position.

2) We will perform a sensitivity analysis for any retinal artery occlusion, using ICD-9-CM codes 362.3, 362.30, 362.31, 362.32, 362.33 and ICD-10-CM codes H34.9, H34.233, H34.1x, H34.0x, H34.12, H34.13, H34.23x, H34.21x) [31].

i) We will perform a secondary analysis of patients with CRAO who later developed atrial fibrillation, to assess the hypothesis that clinically undetected paroxysmal AF may be associated with CRAO.

j) Table shell for Aim 1 regression results are provided in Tables 6 and 7. The primary outcome is underlined.

k) To address confounding resulting from patients potentially receiving a competing therapy (left atrial appendage occlusion, a procedural alternative to long-term oral anticoagulation), we will exclude patients with ICD-9-CM procedure code 37.90 or ICD-10-CM 02L73DK either in the follow-up period or in the ascertainment period.

We will report a figure showing the adjusted cumulative incidence of CRAO according to AF vs. No AF groups using an overlap-weighted cumulative incidence function.

**Regression modeling (aim 2).**

a) In keeping with target trial emulation methodology as described above, the index date for patients exposed to anticoagulation will be the date of the first anticoagulant prescription

**Table 6. Crude and adjusted rates of CRAO in patients with and without atrial fibrillation.**

| | AF | | No AF | | AF | No AF | Rate Difference (95% CI) | aHR (95% CI) |
|---|---|---|---|---|---|---|---|---|
| | Patients with event | Rate per 1000 Or 100,000 Person-years | Patients with event | Rate per 1000 Or 100,000 Person-years | Adjusted rate per 1,000 person-years | Adjusted rate per 1,000 person-years | | |
| CRAO (Overlap weighted) | | | | | | | | |
| CRAO (primary Dx) | | | | | | | | |
| Any RAO | | | | | | | | |
| Ischemic stroke | | | | | | | | |
| UTI | | | | | | | | |

drug claim occurring after the diagnosis of AF. For potential controls, only patients with no exposure to anticoagulation throughout the study period would be eligible to avoid selecting controls who start an anticoagulant during the follow-up period. For controls, inclusion and exclusion criteria will be examined for each year after AF diagnosis, and the midpoint of one random year will be selected as the index date for each control patient.

b) Similar to Aim 1, patients diagnosed with CRAO precedent to or concurrent to AF diagnosis will be excluded and these patients will be examined in an exploratory analysis.

c) Our primary analysis will utilize propensity score overlap weighting methodology to approximate a randomized design where baseline hazard of CRAO of patients receiving anticoagulation vs. not receiving is comparable. We will incorporate the following variables to define the propensity score: age, time since first AF claim, sex, geography, race, Medicaid dual eligibility, vascular comorbidities and non-vascular comorbidities relevant to the likelihood that a patient will be treated with anticoagulation (to include all of the comorbidity variables included for Aim 1 propensity score, as well as history of gastrointestinal bleeding or peptic ulcer disease (using validated ICD-9-CM codes 531.x (gastric ulcer), 532.x (duodenal ulcer), and 578.x (gastrointestinal hemorrhage) and their ICD-10-CM counterparts K25.x, K26.x, and K92.2) and history of intracranial hemorrhage (any of ICD-10-CM I60, I61, S06.3, S06.4, S06.5, S06.6 and their ICD-9-CM counterparts 430.x, 431.x, 432.x)) [32–35].

d) Our principal effect of interest is the hazard of CRAO among patients treated with any oral anticoagulant agent compared to those not treated. Given variable patient adherence to

**Table 7. Hazard ratios for atrial fibrillation vs. no atrial fibrillation for Aim 1.**

| Outcome | Unadjusted Model | Adjusted Model† |
|---|---|---|
| CRAO (Overlap weighted) | HR (95% CI) | aHR (95% CI) |
| CRAO (primary dx) | HR (95% CI) | aHR (95% CI) |
| Any RAO | HR (95% CI) | aHR (95% CI) |
| CRAO (any dx position) | HR (95% CI) | aHR (95% CI) |
| Ischemic Stroke | HR (95% CI) | aHR (95% CI) |
| UTI | HR (95% CI) | aHR (95% CI) |

†This column reflects the results of a model that incorporates overlap weighting using propensity scores re-estimated after our first stage matching procedure.

**Table 8. Results of cause-specific hazard regression modeling for Aim 2, hazard of outcome based on time-varying exposure to anticoagulation.**

| Outcome | Unadjusted Model | Adjusted Model† |
|---|---|---|
| CRAO (Overlap weighted) | HR (95% CI) | aHR (95% CI) |
| CRAO (primary dx) | HR (95% CI) | aHR (95% CI) |
| Any RAO | HR (95% CI) | aHR (95% CI) |
| CRAO (any dx position) | HR (95% CI) | aHR (95% CI) |
| Intracranial Hemorrhage | HR (95% CI) | aHR (95% CI) |
| Ischemic Stroke | HR (95% CI) | aHR (95% CI) |
| UTI | HR (95% CI) | aHR (95% CI) |

†This column reflects the results of a model that incorporates overlap weighting using propensity scores re-estimated after our first stage matching procedure.

anticoagulation, we will take advantage of the strength of Part D data (which records pharmaceutical fills rather than prescriptions) and treat anticoagulation exposure as a time-varying covariate for patients exposed to anticoagulation.

e) We will perform identical sensitivity analyses as described in Aim 1.

f) Table shells for the results of Aim 2 are shown in Table 8.

**Data management plan.** In accordance with Centers for Medicare and Medicaid Services (CMS) regulations, all data will be kept in a secure, Federal Information Security Management Act of 2002 (FISMA) compliant environment managed by the Duke University Department of Population Health Sciences DataShare program. The investigators responsible for drafting the manuscript and preparing dissemination (JL and BMG) will not have access to the raw data or the analytical environment.

**Status and timeline of the study.** Preliminary cohort creation and data management will begin upon submission of this study protocol. Primary statistical analyses will be performed after publication of this protocol, to be completed by the end of calendar year 2023.

**Analytic software.** All analyses will be performed using SAS Version 9.4 (Cary, NC).

# Discussion

## Dissemination plan

Upon conclusion of the study, results will be submitted for publication in a peer-reviewed medical journal. If feasible according to timeline for statistical analysis and abstract submission results will be disseminated at the International Stroke Conference 2024, American Academy of Neurology Annual Meeting 2024, and/or World Stroke Congress 2024.

## Limitations

The principal limitation of this study is its observational, retrospective nature. Unfortunately, the incidence of CRAO is low, and evaluating the effect of anticoagulation on primary or recurrent CRAO risk is likely to be infeasible given the massive study sizes that would be required to detect an effect. While our target trial emulation methodology, including careful selection of index date and control patients and our use of overlap weighting methodology, can help to answer this question to its best extent using real-world data, the potential for

residual confounding is still present in any observational data source. Additional limitations include the use of Medicare data, which has several intrinsic limitations, including having a limited number of beneficiaries under the age of 65, lacking access to clinical data not captured in structured claims, and limited ascertainment of key information on demographic identity, including gender identity and race/ethnicity. Another limitation in any non-controlled study of anticoagulation is the fact that the antithrombotic effect of anticoagulation varies according to patient adherence, especially for patients taking Warfarin. The use of Medicare Part D data has one major advantage in that it captures medication fills, not medication prescriptions, therefore giving a better surrogate of patient adherence, although some patients may not take medications that they nonetheless fill, recognizing that generalizability may be limited to Part D enrollees. Treating exposure to anticoagulation as a time-varying covariate will also help to capture more precisely the effect that therapeutic anticoagulation has on CRAO risk. Furthermore, our use of both positive and negative control endpoints in this analysis should help to interpret our eventual findings: existing evidence from randomized controlled trials has shown that use of anticoagulation is highly effective in preventing embolic stroke due to atrial fibrillation, and is associated with an increased risk of intracranial hemorrhage; if we replicate these results in our study design, it gives increased confidence that our results concerning CRAO may be valid. Finally, some patients with contraindication to oral anticoagulation may receive alternative management strategies that can decrease their ischemic stroke risk, such as receipt of a left atrial appendage occlusion device. We address this limitation through a sensitivity analysis excluding any patient who receives a LAAO during the study follow-up period or with a diagnostic code indicating the placement of an LAAO in the comorbidity ascertainment window.

## Supporting information

**S1 Fig. Directional acyclic graph for Aim 1.** This directional acyclic graph shows our approach to addressing confounding for our first aim.
(TIF)

**S2 Fig. Directional acyclic graph for Aim 2.** This directional acyclic graph shows our approach to addressing confounding for our second aim.
(TIF)

## Author Contributions

**Conceptualization:** Jay B. Lusk, Lauren Wilson, Vinit Nalwade, Ailin Song, Matthew Schrag, Valerie Biousse, Fan Li, Sven Poli, Jonathan Piccini, Ying Xian, Emily O'Brien, Brian Mac Grory.

**Writing – original draft:** Jay B. Lusk, Brian Mac Grory.

**Writing – review & editing:** Lauren Wilson, Vinit Nalwade, Ailin Song, Matthew Schrag, Valerie Biousse, Fan Li, Sven Poli, Jonathan Piccini, Ying Xian, Emily O'Brien.

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
