## [Decision Letter · Decision Letter 0]

10 Oct 2023

PONE-D-23-12441Atrial Fibrillation as a Novel Risk Factor for Retinal Stroke: A Protocol for A Population-Based Retrospective Cohort StudyPLOS ONE

Dear Dr. Mac Grory,

Thank you for submitting your manuscript to PLOS ONE. After careful consideration, we feel that it has merit but does not fully meet PLOS ONE’s publication criteria as it currently stands. Therefore, we invite you to submit a revised version of the manuscript that addresses the points raised during the review process.

We have invited 4 reviewers Please kindly address the concerns of the 4th reviewer==============================

We look forward to receiving your revised manuscript.

Kind regards,

Yee Gary Ang, MBBS MPH

Academic Editor

PLOS ONE

““I have read the journal's policy and the authors of the manuscript have the following interests: J.B. Lusk is supported by the American Heart Association. Dr Biousse is supported by the National Institutes of Health National Eye Institute core grant P30-EY06360 (Department of Ophthalmology, Emory University School of Medicine) and by a departmental grant from the Research to Prevent Blindness (New York, NY) and is a consultant for GenSight Biologics and Neurophoenix. Dr Schrag reports compensation from the REVISION trial for data and safety monitoring services; compensation from Raymond James for consultant services; grants from the National Institutes of Health; and compensation from Labaton Sucharow for consultant services. Dr Poli reports grants from the Helena Laboratories Corporation, Bristol Myers Squibb, Boehringer Ingelheim, and the Daiichi Sankyo Company; compensation from Alexion Pharmaceuticals, Inc, Portola Pharmaceuticals LLC, Werfen USA LLC, and AstraZeneca for consultant services; and compensation from Boehringer Ingelheim, Bristol Myers Squibb, the Daiichi Sankyo Company, and Bayer Healthcare for other services. Dr Piccini reports compensation from Biotronik, Inc, Philips, Medtronic, Boston Scientific Corporation, ResMed Foundation, Ablacon, LivaNova USA, Inc, AltaThera Pharmaceuticals LLC, AbbVie Biotherapeutics, Abbott Laboratories, ElectroPhysiology Frontiers, Itamar Medical, Inc, ARCA biopharma, Pfizer, Bristol Myers Squibb, Sanofi US Services, Inc, and Milestone Scientific, for consultant services; employment by the Duke University; and grants from the American Heart Association and iRhythm Technologies. Dr O’Brien reports compensation from Boehringer Ingelheim for consultant services and employment by the Duke University. B. Mac Grory is supported by a grant from the National Institutes of Health (K23HL161426). The other authors report no conflicts.”

3. Please upload a copy of Figure 1, to which you refer in your text on page 16. If the figure is no longer to be included as part of the submission please remove all reference to it within the text.

Reviewers' comments:

Reviewer's Responses to Questions

**Comments to the Author**

1. Does the manuscript provide a valid rationale for the proposed study, with clearly identified and justified research questions?

Reviewer #1: Yes

Reviewer #2: Yes

Reviewer #3: Yes

Reviewer #4: Yes

2. Is the protocol technically sound and planned in a manner that will lead to a meaningful outcome and allow testing the stated hypotheses?

Reviewer #1: Yes

Reviewer #2: Yes

Reviewer #3: Yes

Reviewer #4: Partly

3. Is the methodology feasible and described in sufficient detail to allow the work to be replicable?

Reviewer #1: Yes

Reviewer #2: Yes

Reviewer #3: Yes

Reviewer #4: No

4. Have the authors described where all data underlying the findings will be made available when the study is complete?

Reviewer #1: Yes

Reviewer #2: Yes

Reviewer #3: Yes

Reviewer #4: No

5. Is the manuscript presented in an intelligible fashion and written in standard English?

Reviewer #1: Yes

Reviewer #2: Yes

Reviewer #3: Yes

Reviewer #4: Yes

6. Review Comments to the Author

You may also provide optional suggestions and comments to authors that they might find helpful in planning their study.

Reviewer #1: I believe that this protocol paper is well written and the research plan is important. Thank you for the opportunity to review your excellent research plan.

Reviewer #2: Dear Authors,

I read your protocol with great interest. I think this will be a very important study, at least partially filling the knowledge gap about CRAO. I do not have major comments on the text of manuscript.

Reviewer #3: Previous study by the corresponding author suggested that the rate of AF detection after CRAO is higher than that seen in age-, sex-, and comorbidity-matched controls and comparable to that seen after ischemic cerebral stroke.[1]

Current study is designed to determine if AF is a risk factor for CRAO independent of other vascular risk factors, and whether use of anticoagulation would modifies the risk of CRAO.

For Aim 1, study design is very comprehensive and included all the vascular risk factors into covariates analysis.

For Aim 2, to compare the hazard of CRAO among patients treated with any oral anticoagulant agent vs those not treated. The group not treated with anticoagulant (probably because of their comorbidity eg: Gastrointestinal bleeding, peptic ulcer disease, intracranial haemorrhage) maybe offered alternative such as Left Atrial Appendage occluder device. The effect of anticoagulation is affected by compliance and dosage. It is stated in the study limitation that there is lack of access to clinical data. We do need to be caution when interpreting the results and drawing conclusions relating to anticoagulation.

In Table 2, symbol * was referring to reference 22 that claims data do not sensitively identify patients with hyperlipidemia. However this symbol also appear under multiple ICD-9 and ICD-10 coding. Please identify if this is intentional or typing error.

Reference:

1. Brian Mac Grory, Sean R. Landman, Paul D. Ziegler, Chantal J. Boisvert, Shane P. Flood, Christoph Stretz, et al. Detection of Atrial Fibrillation After Central Retinal Artery Occlusion. Stroke. 2021;52:2773–2781. Available from https://doi.org/10.1161/STROKEAHA.120.033934

Reviewer #4: # General comments

Although the study claims to be studying a risk factor, in practice, the authors are trying to establish some causal relationships between the risk factor (AF) and the outcome (retinal stroke). This can be seen from the use of propensity scores whose main purpose is to form two or more similar groups for comparisons in order to generate the evidence for causal relationships as well as mentions of trial emulation methodology.

# Adjustment covariates

Under the context of this being a study aimed at estimating causal effects, a causal diagram is required. This causal diagram needs to include all the outcomes of interests and the variables used for adjustment to make the case that the adjustments are appropriate. Some of the covariates appear to lie on the causal path between AF and retinal stroke and a causal diagram would be able to clarify that. Variables lying on the causal pathway should not be adjusted for.

# Propensity score matching

It is unclear how the propensity scores are to be estimated and how they are to be used. What weighting scheme is used and why? Are there risks to under-representation of groups of patients that do not have matched samples and if there are, what is done to mitigate this?

Are there any attempts at assessing unmeasured confounding (treatment - outcome confounding) using sensitivity analyses or other methods? This needs to be done in order to have a reasonable assessment of the robustness of your results.

# Regression modeling

- How do you intend to analyse CRAO over time and across states? No details are offered on this.

- Again, the adjustment for covariates needs to be justified using a causal diagram.

- Is the sequential adjustment an attempt to identify a best-fit model? If not please explain what you mean by sequential adjustment.

7. PLOS authors have the option to publish the peer review history of their article (what does this mean?). If published, this will include your full peer review and any attached files.

Reviewer #1: **Yes: **Kanta Tanaka

Reviewer #2: No

Reviewer #3: No

Reviewer #4: No

---

## [Author Response · Author response to Decision Letter 0]

22 Nov 2023

We have revised our manuscript to adhere to the PLOS ONE style template. 

““I have read the journal's policy and the authors of the manuscript have the following interests: J.B. Lusk is supported by the American Heart Association. Dr Biousse is supported by the National Institutes of Health National Eye Institute core grant P30-EY06360 (Department of Ophthalmology, Emory University School of Medicine) and by a departmental grant from the Research to Prevent Blindness (New York, NY) and is a consultant for GenSight Biologics and Neurophoenix. Dr Schrag reports compensation from the REVISION trial for data and safety monitoring services; compensation from Raymond James for consultant services; grants from the National Institutes of Health; and compensation from Labaton Sucharow for consultant services. Dr Poli reports grants from the Helena Laboratories Corporation, Bristol Myers Squibb, Boehringer Ingelheim, and the Daiichi Sankyo Company; compensation from Alexion Pharmaceuticals, Inc, Portola Pharmaceuticals LLC, Werfen USA LLC, and AstraZeneca for consultant services; and compensation from Boehringer Ingelheim, Bristol Myers Squibb, the Daiichi Sankyo Company, and Bayer Healthcare for other services. Dr Piccini reports compensation from Biotronik, Inc, Philips, Medtronic, Boston Scientific Corporation, ResMed Foundation, Ablacon, LivaNova USA, Inc, AltaThera Pharmaceuticals LLC, AbbVie Biotherapeutics, Abbott Laboratories, ElectroPhysiology Frontiers, Itamar Medical, Inc, ARCA biopharma, Pfizer, Bristol Myers Squibb, Sanofi US Services, Inc, and Milestone Scientific, for consultant services; employment by the Duke University; and grants from the American Heart Association and iRhythm Technologies. Dr O’Brien reports compensation from Boehringer Ingelheim for consultant services and employment by the Duke University. B. Mac Grory is supported by a grant from the National Institutes of Health (K23HL161426). The other authors report no conflicts.”

Please see our updated competing interests statement:

“I have read the journal's policy and the authors of the manuscript have the following interests: J.B. Lusk is supported by the American Heart Association. Dr Biousse is supported by the National Institutes of Health National Eye Institute core grant P30-EY06360 (Department of Ophthalmology, Emory University School of Medicine) and by a departmental grant from the Research to Prevent Blindness (New York, NY) and is a consultant for GenSight Biologics and Neurophoenix. Dr Schrag reports compensation from the REVISION trial for data and safety monitoring services; compensation from Raymond James for consultant services; grants from the National Institutes of Health; and compensation from Labaton Sucharow for consultant services. Dr Poli reports grants from the Helena Laboratories Corporation, Bristol Myers Squibb, Boehringer Ingelheim, and the Daiichi Sankyo Company; compensation from Alexion Pharmaceuticals, Inc, Portola Pharmaceuticals LLC, Werfen USA LLC, and AstraZeneca for consultant services; and compensation from Boehringer Ingelheim, Bristol Myers Squibb, the Daiichi Sankyo Company, and Bayer Healthcare for other services. Dr Piccini reports compensation from Biotronik, Inc, Philips, Medtronic, Boston Scientific Corporation, ResMed Foundation, Ablacon, LivaNova USA, Inc, AltaThera Pharmaceuticals LLC, AbbVie Biotherapeutics, Abbott Laboratories, ElectroPhysiology Frontiers, Itamar Medical, Inc, ARCA biopharma, Pfizer, Bristol Myers Squibb, Sanofi US Services, Inc, and Milestone Scientific, for consultant services; employment by the Duke University; and grants from the American Heart Association and iRhythm Technologies. Dr O’Brien reports compensation from Boehringer Ingelheim for consultant services and employment by the Duke University. B. Mac Grory is supported by a grant from the National Institutes of Health (K23HL161426). The other authors report no conflicts. This does not alter our adherence to PLOS ONE policies on sharing data and materials.”

3. Please upload a copy of Figure 1, to which you refer in your text on page 16. If the figure is no longer to be included as part of the submission please remove all reference to it within the text.

We apologize for the confusion. We do not intend to submit an actual figure, but are simply declaring in our statistical analysis plan that we will report a figure. We have removed the word “figure 1” to make this clear. 

 

REVIEWER COMMENTS

Reviewer #1: I believe that this protocol paper is well written and the research plan is important. Thank you for the opportunity to review your excellent research plan.

We thank the reviewer for the very kind comments about our manuscript. 

 

Reviewer #2: Dear Authors, I read your protocol with great interest. I think this will be a very important study, at least partially filling the knowledge gap about CRAO. I do not have major comments on the text of manuscript.

We thank the reviewer for their positive review of our work. 

 

Reviewer #3: Previous study by the corresponding author suggested that the rate of AF detection after CRAO is higher than that seen in age-, sex-, and comorbidity-matched controls and comparable to that seen after ischemic cerebral stroke.[1] Current study is designed to determine if AF is a risk factor for CRAO independent of other vascular risk factors, and whether use of anticoagulation would modifies the risk of CRAO.

For Aim 1, study design is very comprehensive and included all the vascular risk factors into covariates analysis.

For Aim 2, to compare the hazard of CRAO among patients treated with any oral anticoagulant agent vs those not treated. The group not treated with anticoagulant (probably because of their comorbidity eg: Gastrointestinal bleeding, peptic ulcer disease, intracranial haemorrhage) maybe offered alternative such as Left Atrial Appendage occluder device. The effect of anticoagulation is affected by compliance and dosage. It is stated in the study limitation that there is lack of access to clinical data. We do need to be caution when interpreting the results and drawing conclusions relating to anticoagulation.

We thank the reviewer for this excellent point. We have added the following discussion of this limitation in the limitations section of our manuscript: 

“Finally, some patients with contraindication to oral anticoagulation may receive alternative management strategies that can decrease their ischemic stroke risk, such as receipt of a left atrial appendage occlusion device. We address this limitation through a sensitivity analysis excluding any patient who receives a LAAO during the study follow-up period or with a diagnostic code indicating the placement of an LAAO in the comorbidity ascertainment window.”

Furthermore, we have added to the methods section that we will perform a sensitivity analysis excluding any patient with a code indicating the presence of a left atrial appendage occlusion device. 

“To address confounding resulting from patients potentially receiving a competing therapy (left atrial appendage occlusion, a procedural alternative to long-term oral anticoagulation), we will exclude patients with ICD-9-CM procedure code 37.90 or ICD-10-CM 02L73DK either in the follow-up period or in the ascertainment period.”

In Table 2, symbol * was referring to reference 22 that claims data do not sensitively identify patients with hyperlipidemia. However this symbol also appear under multiple ICD-9 and ICD-10 coding. Please identify if this is intentional or typing error.

We appreciate this excellent catch from the reviewer. We erroneously used asterisks both to denote wild-card fields in ICD-9 and ICD-10 code lists as well as for footnotes in the table. We have replaced the asterisks for footnotes with the dagger (†) symbol to avoid confusion. 

Reference:

1. Brian Mac Grory, Sean R. Landman, Paul D. Ziegler, Chantal J. Boisvert, Shane P. Flood, Christoph Stretz, et al. Detection of Atrial Fibrillation After Central Retinal Artery Occlusion. Stroke. 2021;52:2773–2781. Available from https://doi.org/10.1161/STROKEAHA.120.033934

Reviewer #4: 

# General comments

Although the study claims to be studying a risk factor, in practice, the authors are trying to establish some causal relationships between the risk factor (AF) and the outcome (retinal stroke). This can be seen from the use of propensity scores whose main purpose is to form two or more similar groups for comparisons in order to generate the evidence for causal relationships as well as mentions of trial emulation methodology.

We agree with the reviewer. While we cannot definitively establish causality in this study, our aim is to determine if there is sufficient evidence that atrial fibrillation may cause retinal stroke. 

# Adjustment covariates

Under the context of this being a study aimed at estimating causal effects, a causal diagram is required. This causal diagram needs to include all the outcomes of interests and the variables used for adjustment to make the case that the adjustments are appropriate. Some of the covariates appear to lie on the causal path between AF and retinal stroke and a causal diagram would be able to clarify that. Variables lying on the causal pathway should not be adjusted for.

We appreciate the reviewer’s important comments. We have added two directed acyclic graphs as supplemental figures 1 and 2 to summarize our approach for the two aims. 

Added to the adjustment covariates subsection, “Supplemental Figures 1 and 2 show the directional acyclic graphs representing our understanding of possible confounding variables.”

# Propensity score matching

It is unclear how the propensity scores are to be estimated and how they are to be used. What weighting scheme is used and why? Are there risks to under-representation of groups of patients that do not have matched samples and if there are, what is done to mitigate this?

We utilize overlap weighting as described on page 12. Overlap weighting is an effective strategy to address the reviewer’s point about under-representation of groups of patients without matched samples as described in reference 27: by utilizing overlap weighting, we effectively minimize the effect of outliers while not excluding patients from the study entirely who, in a traditional propensity score matching design, would have gone unmatched. 

Are there any attempts at assessing unmeasured confounding (treatment - outcome confounding) using sensitivity analyses or other methods? This needs to be done in order to have a reasonable assessment of the robustness of your results.

We appreciate this excellent comment. We have adopted three strategies (as described on page 15) to assess confounding in tandem: 1) performing sensitivity analyses for outcome ascertainment and exposure ascertainment, as described on page 15; 2) Performing a sensitivity analysis of patients with CRAO who later developed atrial fibrillation (not included in our primary analysis) to assess the hypothesis that clinically undetected paroxysmal AF may cause CRAO, which could confound our treatment-outcome association. 3) Use of negative and positive control endpoints, which assess whether unmeasured confounding due to general medical risk could underlie our association. 

# Regression modeling

- How do you intend to analyse CRAO over time and across states? No details are offered on this.

We have provided additional detail on this on page 5. 

“This data source allows us to follow unique patients longitudinally for the duration of their fee-for-service enrollment, regardless of state of residence.”

- Again, the adjustment for covariates needs to be justified using a causal diagram.

Per our prior response, we have added supplemental figures 1 and 2 to the paper. 

- Is the sequential adjustment an attempt to identify a best-fit model? If not please explain what you mean by sequential adjustment.

We appreciate the opportunity to clarify the intent of our sequential adjustment approach. The intent of this approach is to provide a rough estimation of how much attributable confounding is addressed through each step of our modeling strategy, to provide more transparency into sources of exposure-outcome confounding that we are able to address in our paper. 

Added to page 15: “We employ this sequential adjustment strategy to provide insight as to sources of measured confounding.”

---

## [Decision Letter · Decision Letter 1]

1 Dec 2023

PONE-D-23-12441R1Atrial Fibrillation as a Novel Risk Factor for Retinal Stroke: A Protocol for A Population-Based Retrospective Cohort StudyPLOS ONE

Dear Dr. Mac Grory,

Thank you for submitting your manuscript to PLOS ONE. After careful consideration, we feel that it has merit but does not fully meet PLOS ONE’s publication criteria as it currently stands. Therefore, we invite you to submit a revised version of the manuscript that addresses the points raised during the review process.

Please make the suggested changes and resubmit.

We look forward to receiving your revised manuscript.

Kind regards,

Yee Gary Ang, MBBS MPH

Academic Editor

PLOS ONE

Journal Requirements:

Reviewers' comments:

Reviewer's Responses to Questions

**Comments to the Author**

1. Does the manuscript provide a valid rationale for the proposed study, with clearly identified and justified research questions?

Reviewer #5: Yes

2. Is the protocol technically sound and planned in a manner that will lead to a meaningful outcome and allow testing the stated hypotheses?

Reviewer #5: Yes

3. Is the methodology feasible and described in sufficient detail to allow the work to be replicable?

Reviewer #5: Yes

4. Have the authors described where all data underlying the findings will be made available when the study is complete?

Reviewer #5: No

5. Is the manuscript presented in an intelligible fashion and written in standard English?

Reviewer #5: Yes

6. Review Comments to the Author

You may also provide optional suggestions and comments to authors that they might find helpful in planning their study.

Reviewer #5: In this study protocol, a retrospective matched controls study is being proposed to determine if prevalent AF is a risk factor for CRAO and if anticoagulant use changes the risk of CRAO.

Minor revisions:

1- Sample size calculation: Indicate the statistical testing method which attains 90% power.

2- Descriptive analysis: Perhaps the statement, “We will summarize count data using . . .” can be clarified by stating, “We will summarize categorical data using . . .”

3- Tables 7 and 8: Clarify that the “Unadjusted Model” reports univariate results whereas the “Adjusted Model” reports multivariate results.

4- Below Table 7: Indicate the statistical method that will be used to graphically displaying the adjusted cumulative incidence of CRAO.

5- Identify the software that will be used for the statistical analysis.

6- Provide details about the characteristics on which the matched controls will be selected.

7- To assist in the review process, add line numbers to the document.

7. PLOS authors have the option to publish the peer review history of their article (what does this mean?). If published, this will include your full peer review and any attached files.

Reviewer #5: No

---

## [Author Response · Author response to Decision Letter 1]

1 Dec 2023

The authorship team would like to thank the scientific reviewer for their thoughtful and constructive comments. These have greatly enriched the manuscript. Please see below our point-by-point responses below.

1- Sample size calculation: Indicate the statistical testing method which attains 90% power.

-Thank you for this comment. We apologize for this oversight. We have addressed with the following addition:

“Power/sample size calculations were made on the basis of an unadjusted Cox proportional hazards regression model.”

Please note that we felt that an unadjusted mode as the basis for our power calculations was appropriate because we intend to first perform a matching procedure and further covariate adjustment is not anticipated to impact our results.

2- Descriptive analysis: Perhaps the statement, “We will summarize count data using . . .” can be clarified by stating, “We will summarize categorical data using . . .”

-We appreciate this comment. I have re-written this section accordingly.

3- Tables 7 and 8: Clarify that the “Unadjusted Model” reports univariate results whereas the “Adjusted Model” reports multivariate results.

-Thank you very much for this comment. We have updated these tables to ensure this point is conveyed:

“†This column reflects the results of a model that incorporates overlap weighting using propensity scores re-estimated after our first stage matching procedure”

4- Below Table 7: Indicate the statistical method that will be used to graphically displaying the adjusted cumulative incidence of CRAO.

-Thank you very much. We have added the statistical method:

“We will report a figure showing the adjusted cumulative incidence of CRAO according to AF vs. No AF groups using an overlap-weighted cumulative incidence function.”

5- Identify the software that will be used for the statistical analysis.

-Thank you for this comment. We have updated this as follows:

“Analytic Software

All analyses will be performed using SAS Version 9.5 (Cary, NC).”

6- Provide details about the characteristics on which the matched controls will be selected.

-Thank you very much for this comment. We have provided the following details about the characteristics on which the matched controls will be selected:

“All remaining beneficiaries will be considered potential controls; inclusion and exclusion criteria will be applied based on an index date of July 1 for each year in the study period; all beneficiaries meeting inclusion/exclusion criteria for each year will be included in the potential control pool. For all potential controls with multiple index dates/years of eligibility, the year they contribute will be selected randomly. A multivariable logistic regression analysis will be conducted using cases and potential controls to create propensity scores for AF status; the final control group will be selected using propensity score matching to the AF cases.”

7- To assist in the review process, add line numbers to the document.

Thank you, we have made this change.

---

## [Decision Letter · Decision Letter 2]

10 Dec 2023

Atrial Fibrillation as a Novel Risk Factor for Retinal Stroke: A Protocol for A Population-Based Retrospective Cohort Study

PONE-D-23-12441R2

Dear Dr. Mac Grory,

We’re pleased to inform you that your manuscript has been judged scientifically suitable for publication and will be formally accepted for publication once it meets all outstanding technical requirements.

Kind regards,

Yee Gary Ang, MBBS MPH

Academic Editor

PLOS ONE

Additional Editor Comments (optional):

Reviewers' comments:

Reviewer's Responses to Questions

**Comments to the Author**

1. Does the manuscript provide a valid rationale for the proposed study, with clearly identified and justified research questions?

Reviewer #5: Yes

2. Is the protocol technically sound and planned in a manner that will lead to a meaningful outcome and allow testing the stated hypotheses?

Reviewer #5: Yes

3. Is the methodology feasible and described in sufficient detail to allow the work to be replicable?

Reviewer #5: Yes

4. Have the authors described where all data underlying the findings will be made available when the study is complete?

Reviewer #5: No

5. Is the manuscript presented in an intelligible fashion and written in standard English?

Reviewer #5: Yes

6. Review Comments to the Author

You may also provide optional suggestions and comments to authors that they might find helpful in planning their study.

Reviewer #5: All comments have been adequately addressed.

7. PLOS authors have the option to publish the peer review history of their article (what does this mean?). If published, this will include your full peer review and any attached files.

Reviewer #5: No

---

## [Editor Report · Acceptance letter]

18 Dec 2023

PONE-D-23-12441R2 

PLOS ONE

Dear Dr. Mac Grory, 

I'm pleased to inform you that your manuscript has been deemed suitable for publication in PLOS ONE. Congratulations! Your manuscript is now being handed over to our production team.

Kind regards, 

on behalf of

Dr. Yee Gary Ang 

Academic Editor

PLOS ONE